# Sputtered Non-Hydrogenated Amorphous Silicon as Alternative Absorber for Silicon Photovoltaic Technology

**DOI:** 10.3390/ma14216550

**Published:** 2021-11-01

**Authors:** Susana Fernández, J. Javier Gandía, Elías Saugar, Mª Belén Gómez-Mancebo, David Canteli, Carlos Molpeceres

**Affiliations:** 1Departamento de Energía, CIEMAT, Avd. Complutense 40, 28040 Madrid, Spain; jj.gandia@ciemat.es (J.J.G.); elias.saugar@ciemat.es (E.S.); 2División de Química, CIEMAT, Avda. Complutense 40, 28040 Madrid, Spain; mariabelen.gomez@ciemat.es; 3Centro Láser, Universidad Politécnica de Madrid-Campus Sur, Alan Turing 1, 28031 Madrid, Spain; david.canteli@upm.es (D.C.); carlos.molpeceres@upm.es (C.M.)

**Keywords:** non-hydrogenated amorphous silicon, alternative light absorbers, magnetron sputtering, low-cost processing, photovoltaic technology

## Abstract

Non-hydrogenated amorphous-silicon films were deposited on glass substrates by Radio Frequency magnetron sputtering with the aim of being used as precursor of a low-cost absorber to replace the conventional silicon absorber in solar cells. Two Serie of samples were deposited varying the substrate temperature and the working gas pressure, ranged from 0.7 to 4.5 Pa. The first Serie was deposited at room temperature, and the second one, at 325 °C. Relatively high deposition rates above 10 Å/s were reached by varying both deposition temperature and working Argon gas pressure to ensure high manufacturing rates. After deposition, the precursor films were treated with a continuous-wave diode laser to achieve a crystallized material considered as the alternative light absorber. Firstly, the structural and optical properties of non-hydrogenated amorphous silicon precursor films were investigated by Raman spectroscopy, atomic force microscopy, X-ray diffraction, reflectance, and transmittance, respectively. Structural changes were observed in the as-deposited films at room temperature, suggesting an orderly structure within an amorphous silicon matrix; meanwhile, the films deposited at higher temperature pointed out an amorphous structure. Lastly, the effect of the precursor material’s deposition conditions, and the laser parameters used in the crystallization process on the quality and properties of the subsequent crystallized material was evaluated. The results showed a strong influence of deposition conditions used in the amorphous silicon precursor.

## 1. Introduction

Silicon is by far the most common material used in solar cells, resulting in a key piece for the photovoltaic industry; it is the second most abundant material on Earth, is non-toxic, and shows high stability and durability [1,2,3,4]. Accordingly, the photovoltaic sector is dominated by wafer-based crystalline silicon (c-Si) and multi-crystalline silicon (mc-Si) solar cells, with a throughput about 20–25%. In fact, 90% of the manufactured solar cells are based on these technologies [5]. Despite the dramatic cost reduction undergone by mono- and mc-Si wafers in the last decade, the further gain in competitiveness of photovoltaic devices requires intensive search for new approaches in what concerns semiconductor devices. The road to find new both solutions and forms of silicon goes through thickness reduction, kerf avoidance, and/or manufacturing cost lowering. Nowadays, the main efforts are focused on searching new cheap alternatives capable to replace above technologies. Although Czochralski (Cz) method is a well-known to obtain high quality c-Si wafer of around 100 μm-thick, it is a costly process that implies a high material consumption [6]. Therefore, the fabrication of thin films of polycrystalline silicon (poly-Si) (≤40 μm) from non-hydrogenated amorphous silicon (a-Si) as precursor layer is very promising [7,8,9]. This poly-Si shows several advantages in comparison with the Cz technique, including the significant saving in material [10]. Liquid phase crystallization (LPC) of silicon is a promising method to achieve high quality thin-film poly-Si from the crystallization of a-Si thin films deposited on glass by using continuous wave diode laser or electron beam [7,11]. LPC technology offers an alternative solution to the well-established Cz method. In this sense, several works have already demonstrated excellent results using LPC silicon absorber on glass, with efficiencies up to 14.8% and short circuit voltages higher than 600 mV [12,13]. However, this silicon precursor is fabricated by high-rate electron beam evaporation or by plasma-enhanced chemical vapor deposition (PECVD) at elevated substrate temperatures as high as 600 °C [14] or using post-deposition treatments at temperatures at temperatures as high as 1000 °C [15], which still implies a high energy consumption.

In the last decades, a-Si is gaining importance due to its advantages versus c-Si: (i) it is more absorbent in the visible wavelength (band gap energy of 1.7 eV vs. 1.1 eV, respectively) and (ii) its absorption coefficient is one order of magnitude higher, leading to a thickness reduction [16]. Hence, its use allows both energy and material saving. Another important advantage with respect to other materials is its low deposition temperature that permits the use of flexible and lightweight substrates over the conventional c-Si. In addition, it is well-known that the growth process and the conditions used in the fabrication of the precursor material could affect the properties of the subsequent laser crystallized material. Thus, it is essential to firstly evaluate the quality and the nature of the precursor material before carrying out the laser process.

Considering the above, this work presents an analysis of the suitability of the non-hydrogenated a-Si precursor layers fabricated by radio frequency (RF) magnetron sputtering at a low-temperature regimen. This deposition technique shows several advantages such as to allow (i) depositing a-Si films from an inexpensive doped silicon (Si) target, (ii) using an inert gas (Ar) to avoid toxic gases as silane, (iii) not using high temperature regimes as techniques such as electron beam, and (iv) achieving deposition rates higher than other deposition methods (i.e., Plasma-Enhanced Chemical Vapor Deposition, PECVD [17]). The effects of the deposition conditions on the a-Si precursor properties (i.e., structure/nature, refractive index and/or band gap energy) and on the subsequent crystallized layers (i.e., grain size, crystalline faction) are studied. We are looking for a precursor material fabricated at moderate temperature (not higher than 325 °C) with a cost-effective technique, without needing high temperature processes, except the unavoidable step of laser treatment, and at fabrication conditions that trigger the LPC. Such an achievement can be considered quite relevant to obtain suitable precursors fabricated at a low energy consumption regime.

## 2. Materials and Methods

The non-hydrogenated a-Si thin films were deposited on glass (3 mm-thick Borofloat^®^ 33 and Corning Eagle XG^®^ coated with a 200 nm-thick SiO_2_, that favors the subsequent crystallization process) substrates using a commercial single chamber magnetron sputtering MVSystem with one cathode, vertically adjustable, operated by RF power. The distance between the target-substrate was set to 6.1 cm. The 3 inch-size diameter p-Si target (Si) came from Lesker Company (St. Leonards-on-Sea, East Sussex, UK)) with a purity of 99.999%, a resistivity of 0.005–0.020 Ω·cm and a density of 2.32 g/cm^3^. The control of the substrate temperature was performed by a type K reference thermocouple. Prior to load the substrates into the chamber, they were cleaned by using ultrasonic baths, rinsed in deionized water, and dried by blown nitrogen. Prior to the sputtering deposition, the base pressure of the chamber was around to 3 × 10^−4^ Pa. The a-Si precursor films were deposited at the different RF powers of 525 W and 450 W, at the substrate temperatures of room temperature (RT) and 325 °C and at the working gas argon (Ar) pressure ranged from 0.7 to 4.5 Pa that corresponded to an Ar flux of 17 to 70 sccm, respectively. These deposition conditions were chosen to obtain the highest possible deposition rate allowed by the sputtering system, and to look for compact layers, in agreement with Thornton model [18].

The crystallization process of the a-Si precursor was carried out with a continuous-wave (CW) diode laser emitting at 940 nm with a spot at focus of 16.5 mm × 2.16 mm. This spot is linear along the X axis and Gaussian in the Y axis. In addition, to reduce the possible thermal stress caused by this process and to prevent the possible resultant film cracking, the a-Si layers were previously heated at the high temperature of 650 °C on a heating table. After that, the samples were moved in a straight line at a constant scanning speed ranged from 400 to 1000 nm/min, and a laser irradiance was varied from 900 J/cm^2^ and 1650 J/cm^2^ were used.

The structural properties and morphology of a-Si films and the subsequent crystallized material were studied by Raman Spectroscopy, X-ray Diffraction (XRD), and Atomic Force Microscopy (AFM), respectively. A deeper morphological study was also carried out by micro-X-ray diffraction (μXRD) to obtain more information about the micro-structural nature of the precursor films. The Raman spectra were measured with a Micro-Raman LabRam HR Evolution (Tulln, Austria) with a HeNe at 632.81 nm laser as excitation source. XRD patterns were collected by using a PANalytical X’Pert Pro diffractometer operating in θ–2θ configuration, with CuKα radiation (45 kV–40 mA), in the angular range of 20° < 2θ < 80°. Phase identification was obtained by comparison with The Inorganic Crystal Structure Database (ICDS). The surface morphology was analyzed by using a standard AFM Multimode Nanoscope III-A SPM from Veeco-Digital Instruments (Cambridgeshire, UK) operating in tapping mode. The roughness was quantified by the Root Mean Square (RMS) deviation of the AFM measured height from the mean data plane in the 5 × 5 µm^2^ images. Images were also taken in smallest areas of 1 × 1 µm^2^.

Finally, the specular transmittance (T) and reflectance (R) spectra of the as-deposited samples were obtained with a Perkin-Elmer Lambda 1050 UV/Visible/NIR spectrophotometer (Waltham, MA, USA), illuminating from the film side. From these data, film thickness (d), refraction index (n), and absorption coefficient (α) were obtained by using a home-software. By means of the Tauc-plot of the absorption coefficient, the energy gap (E_G_) was extracted.

## 3. Results

### 3.1. Sputtering Deposition of Low-Temperature Non-Hydrogenated Amorphous Silicon (Precursor)

Two set of samples of ~1.2 μm-thick samples were deposited at RT and 525 W of RF power (Series A), and at 325 °C and 450 W of RF power (Series B), varying working gas pressure from 0.7 to 4.5 Pa. Table 1 described the conditions used in the samples in studied.

The deposition conditions were optimized to reach deposition rates above 10 Å/s. Figure 1 shows the dependence of the deposition rate with the working Ar pressure for the Series A and B, depicted by red filled circles and black filled square, respectively.

As it can be observed, the deposition rate linearly decreased with the Ar pressure in both Series, being, in the most cases, superior to 10 Å/s. This value was higher than that achieved with other depositions methods such as PECVD [16]. The decreasing behavior of that parameter was expected due to the reduction in the mean free path of the species in the plasma when the gas pressure raised. The main mechanism responsible of that behavior was the increase in the collisions between the sputtered particles and the charged Ar ions [19].

The structural properties of the deposited a-Si films were investigated by Raman spectroscopy. Figure 2 depicts the Raman spectra of the samples of (a) Series A and (b) Series B.

These spectra were fitted at four main bands [20,21]: A broad band centered at the wavenumber of ∼480 cm^−1^ that corresponded to an amorphous nature, that one at the wavenumbers of 500–510 cm^−1^, related to a nano-crystalline (nc) structure, the band at 510–520 cm^−1^, related to a polycrystalline (pc) material, and the last one at 520 cm^−1^, corresponding to the single crystal silicon.

Raman spectra of Series A samples revealed a structural transition from nanocrystalline to amorphous nature as the Ar pressure increased. At the low Ar pressure regimen used, the Si–Si transverse optical mode (TO) was located at 500 cm^−1^, corresponded to nc-Si. This nc nature could be attributed to the higher impact of the sputtered ions on the substrate surface yielded at low Ar pressures [19]. In addition, the films deposited at these conditions of RT and low pressures would show a compact structure die to the high deposition rate, favoring to a better structural quality and to the presence of such nc structures. However, as the Ar pressure raised, more collisions were promoted in the plasma between the energetic Ar ions and the sputtered atoms that would lose part of its energy impacting much softer on the surface substrate [22]. Thus, the atoms would have lower kinetic energy, not enough to achieve a crystalline structure. To determine semi quantitatively the crystalline volume fraction $X_{C},\;$the ratio of the integrated scattering intensity of the crystalline phase to the total scattering intensity was calculated from Raman spectra, according with the following equation [23]:(1)$$X_{C}=\frac{I_{500-510}}{\left(I_{500-510}+I_{480}\right)},$$
where *I* is the area of the Gaussian peaks fitted to the spectra. The $X_{C}$ values calculated were 20% and 17% for the samples A1 and A2, respectively.

On the other hand, the Raman spectra of Series B samples showed a completely amorphous nature, where the Si–Si TO appeared at the wavenumber band of 470–480 cm^−1^. This behavior goes against what has been observed in the literature, where the films deposited at high temperatures are denser, with an enhanced structural quality, and smoother surfaces due to both the high kinetic energy and the high surface diffusion [24]. A possible explanation for that unusual behavior would be that the samples of Series B were deposited at lower deposition rate and RF powers than samples of Series A. Hence, although the sputtered atoms in Series B would have enhanced surface diffusion due to the high substrate temperature, this would not be large enough to form an nc phase and to compensate the difference in the kinetic energy, as it would be happening in the samples of Series A.

To corroborate the structural and morphological difference between both as-deposited series, a deeper study using micro X-ray diffraction (μXRD) was carried out. This technique permits to observe slight changes that are not possible with other configurations. Representative XRD patterns of Series A and B are shown in Figure 3.

Despite the deposition of Series B was carried out at relatively high temperature, the patterns (Figure 3b) did not present significant peaks. The broad band at ~25° was attributed to an amorphous structure [21]. Other authors as Cheng et al. and Jun et al. [25,26] reported an improvement of the crystallinity with increasing the substrate temperature from RT (completely amorphous) to 900 °C ((111), (220), and (331) crystal planes of silicon), although in these works the RF power applied (1200 W) [23] or the tested temperatures (from 600 to 900 °C) [26] were completely differed from the used ones in this work. This may be due to the combination of RF power and substrate temperature (450 W and 325 °C) that could not be high enough to improve the crystallinity of the as-deposited samples, and hence, the increase in the working pressure would not have any observable effect on these samples. However, in the XRD patterns of the samples of Series A (samples A1 and A2) (Figure 3a), three pronounced diffraction peaks can be noticed at 2θ ≅ 45.3°, 65.6°, and 78.6°, which corresponded to SiO_2_ peaks, and (400) and (331), to crystal planes of silicon [24], indicative of a possible ordered structure embedded in an amorphous matrix. The SiO_2_ peak was attributed to the surface native oxide, which is probably presented but masked in the spectra of the amorphous films. Y. Qin et al. [27] reported previously that the samples deposited at RT and at different RF powers no present significant XRD peaks. However, in this work, it is possible to see an evolution from amorphous to crystallinity by the effect of pressure, although this effect was not observed in the sample at higher pressure (A3), probably as the pressure was too high.

Figure 4 depicts the 2D AFM micrographs of samples of (a) Series A and (b) Series B. A clear mounded morphology with an appreciable coarsening process was observed on the surface of the amorphous samples, resulting more evident in the samples deposited at 325 °C. On the other hand, spherical and uniform grains on the surface of the samples A1 and A2 were observed.

Table 2 summarizes the RMS values for both Series measured in the 5 × 5 µm^2^ AFM micrographs. The results indicated that smoother surfaces were obtained at low working pressures and at high substrate temperature (Series B). The change in the RMS values could be explained as the working pressure increased; the films tended to be rougher, led to a columnar microstructure, in agreement with the Thornton’s structure zone model (SZM) [18].

Regarding the optical parameters of a-Si films, the electronic structure and band tails make the E_G_ not have an accurate definition but having a value close to 1.7 eV for the hydrogenated material (a-Si: H) [21] and 1.5 eV for non-hydrogenated one [26]. To define the optical band gap transition of amorphous semiconductors between valence and conduction bands can be used the Tauc’s equation [28].
(2)$${\left(\alpha E\right)}^{1/2}=B\left(E-E_{G}\right),$$
where *α* is the absorption coefficient, *E*, the photon energy and *B*, a constant called band tailing parameter. Table 3 shows the optical parameters calculated such as the band gap energy *E_G_*, obtained from equation (1), and the refractive index at 1 eV, for the samples of both Series.

A clear blue-shift in E_G_ was observed for the samples of Series A with increasing the working pressure. This difference could be attributed to a change in the film structure, i.e., a possible ordered structure embedded in an amorphous matrix. In addition, at high working Ar pressures, the self-shadowing effects due to the lower mean free path due to the greater number of collisions produced in the plasma during the deposition, were more evident [29]. This would lead to the formation of more porous films, as demonstrated the lower value of the refractive index of the sample A3, of 3.17 in comparison to that calculated for samples A1 and A2. This would be consistent with the greater compactness derived from the presence of nanocrystals within an amorphous matrix, as the XRD and Raman spectra seem to suggest for those samples. Contrarily, no clear trend was observed in the E_G_ and in the refractive index of the samples of Series B, in agreement with the amorphous nature and similar morphology observed for all these samples.

At first sight, it could be thought that starting from an nc raw material could be advantageous for the subsequent crystallized material, and the nanocrystals could be acting as a seed. Taking this into account, the effects of (i) the substrate temperature, and (ii) the initial structure of the a-Si precursor on both the crystallization process and the main parameters of the crystallized material (i.e., crystalline fraction and grain size) are investigated in the next section.

### 3.2. Effect of Deposition Conditions and Structure of the Precursor Material on the Subsequent Crystallized a-Si

Table 4 describes the main parameters used in the crystallization process (i.e., irradiance and scan speed) on the precursor samples A1 and B2, with nanocrystalline and amorphous structure, respectively. The crystalline fraction $X_{C}$ was obtained from Raman spectra, using formula (1), and the grain size was calculated from the XRD patterns of the crystallized samples.

The results reveal that the $X_{C}$ values for the crystallized zones of sample B2, deposited at the moderate temperature of 325 °C, were approximately 20% higher than for sample A1, deposited at RT. Note that B2 supported higher irradiance values during the crystallization process than A1 (see Table 4). In fact, sample A1 did not endure such high irradiance values, and as consequence, the molten material was not sticking to the substrate, suffering a strong peel-off and showing crystallized areas seriously damage. Therefore, the highest crystalline fraction of 84% obtained from A1 was not enough and the LPC was not reached. In addition, the grain size scarcely changed neither with the irradiance nor with the scan speed in the crystallized material using A1 as precursor, even though the raw material presented an nc structure embedded into an amorphous matrix. This limitation observed in the crystallization conditions applied on A1 was attributed to the low deposition temperature, having more weight than the fact of starting from a slight orderly structure. Such a result would be consistent with the finding that from raw material deposited at temperatures higher than RT led in a more effective way to the rearrangement of silicon atoms during the crystallization process, and hence, the LPC can be achieved more easily [12]. Furthermore, it can be observed that the grain size increased sharply from 70 nm to ~2.5 µm for the crystallized material from sample B2, reaching values of crystalline fraction higher than 95%. This was consistent with the higher irradiance values allowed in the crystallization process of this sample.

In view of the results obtained, it can be concluded that the substrate temperature at which the raw material was deposited would have strong influence on the subsequent crystallization parameters used, and hence, on the capability of reaching the LPC. In this work, it has been demonstrated that high $X_{C}\;$ of 95% and grain size of the order of microns can be achieved from a completely amorphous raw material deposited at moderately high deposition temperature of 325 °C, which is still below what is used in other studies [12,13,14].

## 4. Conclusions

a-Si films were deposited by RF magnetron sputtering on glass substrates at different temperatures of RT and 325 °C and working gas pressures ranged from 0.7 to 4.5 Pa. Under these conditions, high deposition rates (>10 Å/s) were reached. This data can be considered as an essential requirement for low-cost photovoltaic technology to fabricate cost-effective absorbers.

The Raman spectra and XRD patterns suggested an nc structure embedded in an amorphous matrix when the precursor samples were deposited at RT and relatively low process pressures up to 3.2 Pa. These samples showed a preliminary crystalline fraction around 20%, and the optical band gap and compactness obtained were closer to a crystalline material than to a purely amorphous one. Lastly, the AFM analysis revealed smoother surfaces when the precursor layers were deposited at the substrate temperature to 325 °C.

On the other hand, the characterization of the crystallized samples showed an improvement of the grain size (~2.5 µm) and the crystalline fraction (94%) when starting from an a-Si precursor material deposited at the moderately high temperature of 325 °C. These results suggest the relevance of the substrate deposition to reach the LPC in the crystallization process, a key piece to achieve a suitable crystallized material. In addition, the laser processed samples offered better performance under high irradiances, regardless the sputtered conditions used in the precursor fabrication. Despite the precursor samples deposited at RT showed an initial nc structure, the values reached of the crystalline fraction and the grain size were very poor, and in any case, not superior to 84% and 71 nm, respectively.

Finally, such an achievement reached using a precursor material deposited at so moderately substrate temperature could result very advantageous for the silicon technology. This fact could be an interesting choice to reduce the energy consumption in the device processing.

## Figures and Tables

**Figure 1 materials-14-06550-f001:**
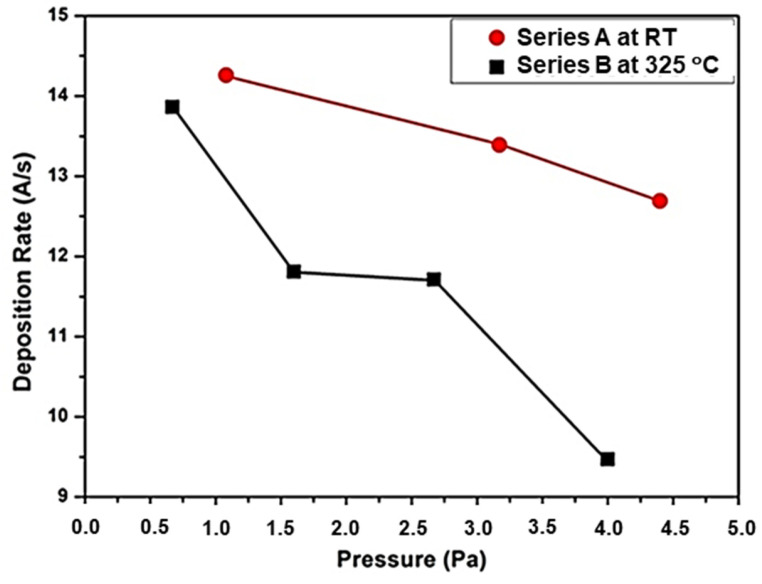
Deposition rate versus working Ar pressure for a-Si films deposited at RT (red symbols) and at 325 °C (black symbols).

**Figure 2 materials-14-06550-f002:**
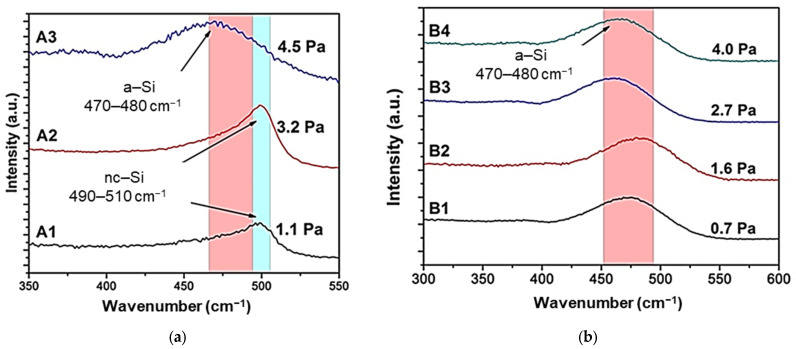
Raman spectra of the samples of: (**a**) Series A deposited at RT; (**b**) Series B deposited at 325 °C.

**Figure 3 materials-14-06550-f003:**
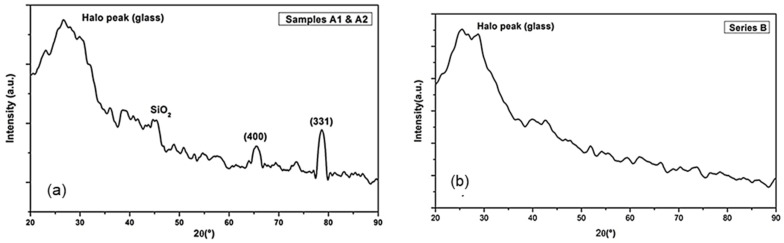
μXRD patterns of samples of Series: (**a**) A deposited at RT; (**b**) Series B deposited at 325 °C.

**Figure 4 materials-14-06550-f004:**
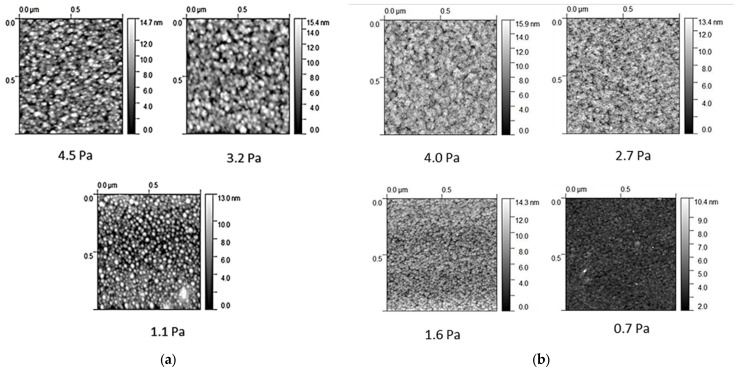
AFM 1 × 1 µm^2^ 2D micrographs of samples of Series: (**a**) A deposited at RT; (**b**) B deposited at 325 °C.

**Table 1 materials-14-06550-t001:** Summarize of the conditions used in the a-Si thin films deposited.

Code Sample	Substrate Temperature (°C)	RF Power (W)	Pressure (Pa)
A 1	RT	525	1.1
A 2	RT	525	3.2
A 3	RT	525	4.5
B 1	325	450	0.7
B 2	325	450	1.6
B 3	325	450	2.7
B 4	325	450	4.0

**Table 2 materials-14-06550-t002:** RMS values of a-Si thin films obtained from 5 × 5 µm^2^ AFM 2D micrographs as a function of the working pressure.

Code Sample	Pressure (Pa)	RMS (nm)
A 1	1.1	1.63
A 2	3.2	1.89
A 3	4.5	1.93
B 1	0.7	0.64
B 2	1.6	0.69
B 3	2.7	1.20
B 4	4.0	1.25

**Table 3 materials-14-06550-t003:** Optical parameters of a-Si thin films as a function of the working pressure.

Code Sample	Pressure (Pa)	*E_G_* (eV)	n @ 1 eV
A 1	1.1	1.18	3.75
A 2	3.2	1.35	3.40
A 3	4.5	1.36	3.17
B 1	0.7	1.42	3.60
B 2	1.6	1.37	3.65
B 3	2.7	1.47	3.70
B 4	4.0	1.48	3.50

**Table 4 materials-14-06550-t004:** Parameters used in the crystallization process of a-Si films, and the crystalline fraction and grain size calculated in the crystallized samples.

Code Sample	Irradiance (W/cm^2^)	Scan Speed (nm/min)	$X_{C}\;(\%)$	Grain Size (nm)
A 1	950	400	84	50
A 1	1220	600	84	71
B 2	1385	1000	91	70
B 2	1630	1000	96	2450

## Data Availability

Not applicable.

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
