# Peer review of "Sputtered Non-Hydrogenated Amorphous Silicon as Alternative Absorber for Silicon Photovoltaic Technology"

_materials, 2021, doi:10.3390/ma14216550_

Round 1

Reviewer 1 Report

The reviewer think the major focus/finding of the paper is to investigate how the characteristics of the precursor a-Si film affects the subsequent crystallization of the film.  The authors have already published 2 papers on the deposition and characterization of RF sputtered a-Si films. 

1) Spectroscopic ellipsometry study of non-hydrogenated fully amorphous silicon films deposited by room-temperature radio-frequency magnetron sputtering on glass: Influence of the argon pressure Márquez , E. Blanco , C. García-Vázquez , J.M. Díaz , E. Saugar, Journal of Non-Crystalline Solids 547 (2020) 120305

2) The influence of Ar pressure on the structure and optical properties of nonhydrogenated a-Si thin films grown by rf magnetron sputtering onto room temperature glass substrates, Márquez , E. Saugar , J.M. Díaz , C. García-Vázquez , S.M. Fernández-Ruano , E. Blanco , J.J. Ruiz-Pérez , D.A. Minkov, Journal of Non-Crystalline Solids 517 (2019) 32–43

In the introduction section, the authors have to discuss their previous papers and clearly explain the novelty of this paper. That will help the readers to understand what exactly is the focus of this paper. 

The reviewer didn't understand why the authors used completely different pressure conditions for A and B series samples. It would have been better if they use similar pressure for both A and B  samples, so that the effect of deposition temperature would be more obvious.

In the XRD plot, figure 3a, authors said it is for samples A1 & A2, but there is only one plot in that. 

There are large number of language-related errors visible throughout the manuscript.  

Author Response

Thank you!

Reviewer 2 Report

The authors presented a work on "Sputtered non-hydrogenated amorphous silicon as alternative absorber for silicon photovoltaic technology ". The work is well written. The method of manufacturing the materials is well described. The samples made are explored with different measurement techniques that allow us to understand the growth and the type of material made. The authors present the results correctly using many figures and references. 

Author Response

Thank you!

Reviewer 3 Report

  1. The paper title is "Sputtered Non-Hydrogenated Amorphous Silicon as Alternative Absorber for Silicon Photovoltaic Technology". However, I cannot find any description or results about using as an absorber. I believed absorber efficiency is the important factor for your films. Authors should add the absorber efficiency in different films.
  2. As I know, depositing the films at higher temperature can improve their crystallization. Please discuss why samples A had better crysallization.
  3. The cross-sectional observation should be added to prove your deposition rates.
  4. Black-and-white images in Figure 4 should be changed to color images. The RMS values in Table 2 cannot match the results in Figure 4.

Author Response

Thank you!

Round 2

Reviewer 3 Report

Accept in present form